# Cohort Study of Antihyperglycemic Medication and Pancreatic Cancer Patients Survival

**DOI:** 10.3390/ijerph17176016

**Published:** 2020-08-19

**Authors:** Audrius Dulskas, Ausvydas Patasius, Donata Linkeviciute-Ulinskiene, Lina Zabuliene, Giedre Smailyte

**Affiliations:** 1Department of Abdominal and General Surgery and Oncology, National Cancer Institute, 1 Santariskiu Str., LT-08406 Vilnius, Lithuania; 2Institute of Clinical Medicine, Faculty of Medicine, Vilnius University, 1 Santariskiu Str., LT-08406 Vilnius, Lithuania; linazabuliene@gmail.com; 3Laboratory of Cancer Epidemiology, National Cancer Institute, LT-08406 Vilnius, Lithuania; ausvydas.patasius@gmail.com (A.P.); giedre.smailyte@nvi.lt (G.S.); 4Institute of Health Sciences, Faculty of Medicine, Vilnius University, LT-03101 Vilnius, Lithuania; 5Institute of Biomedical Sciences, Faculty of Medicine, Vilnius University, LT-03101 Vilnius, Lithuania; linkeviciutei@gmail.com

**Keywords:** pancreatic cancer, metformin, population-based study, antihyperglycemic medications

## Abstract

Background: We assessed the association between the use of metformin and other antihyperglycemic medications on overall survival in diabetic patients with pancreatic cancer. Methods: Patients with pancreatic cancer and diabetes between 2000 and 2015 were identified from the Lithuanian Cancer Registry and the National Health Insurance Fund database. Cohort members were classified into six groups according to type 2 diabetes mellitus treatment: sulfonylurea monotherapy; metformin monotherapy; insulin monotherapy; metformin and sulfonylurea combination; metformin and other antihyperglycemic medications; all other combinations of oral antihyperglycemic medications. Survival was calculated from the date of cancer diagnosis to the date of death or the end of follow-up (31 December 2018). Results: Study group included 454 diabetic patients with pancreatic cancer. We found no statistically significant differences in overall survival between patients by glucose-lowering therapy. However, highest mortality risk was observed in patients on insulin monotherapy, and better survival was observed in the groups of patients using antihyperglycemic medication combinations, metformin alone, and metformin in combination with sulfonylurea. Analysis by cumulative dose of metformin showed significantly lower mortality risk in the highest cumulative dose category (HR 0.76, 95% CI 0.58–0.99). Conclusions: Our study showed that metformin might have a survival benefit for pancreatic cancer patients, suggesting a potentially available option for the treatment.

## 1. Introduction

Pancreatic cancer is considered one of the deadliest cancers worldwide, with few treatment options that have not changed much over the years. Surgery and some forms of chemotherapy give the mean five-year survival of less than 5 to 10% [1]. Therefore, the importance of continuing the search for new possibilities that could affect the course of this disease is unquestionable. Pancreatic cancer is associated with some risk factors: cigarette smoking, fatty diet, diabetes, obesity, alcohol consumption, gender, *Helicobacter pylori* infection, and low physical activity [2].

The International Diabetes Federation (IDF) estimated that 1 in 11 adults aged 20–79 years (463 million of adults) had diabetes mellitus globally in 2019 [3].

Metformin is the most common and most widely prescribed first-line treatment for type 2 diabetes mellitus (T2DM). T2DM can be treated with monotherapy or with a combination of various antihyperglycemic medications such as metformin, sulfonylurea, insulin, or others. Metformin decreases glucose levels by lowering insulin resistance by suppression of hepatic gluconeogenesis and increasing absorption of glucose to peripheral tissues [4]. 

Some in vitro and in vivo experiments have shown metformin to have anticancer effects on pancreatic cancer [5,6,7,8,9]. Metformin has the direct growth-inhibitory action on cancer cells [10] and indirect action by decreasing insulin and insulin grow factor 1 (IGF-1) levels [11]. Many studies have focused on metformin action to inhibit the mechanic target of rapamycin complex 1 (mTORC1) pathway by AMP-activated protein kinase (AMPK) and other mechanisms [12,13,14,15,16,17]. In addition, other investigators suggested different potential antitumorigenic effects [18,19,20,21,22].

Recently, metformin use has been associated with an improved survival of specific cancers, including colorectal, gastric, breast, and prostate cancers [23,24,25,26]. The effect of metformin on pancreatic cancer survival remains controversial. Some studies showed that metformin might improve survival in pancreatic cancer patients [27,28]. However, other studies did not observe significantly better survival in patients who used metformin and underwent surgery, and suggested that the statistical methods were not appropriate in other studies [29]. In addition, two randomized controlled trials recently failed to show significant survival changes when adding metformin to pancreatic cancer treatment [30,31]. 

The aim of this study was to assess the association between the use of metformin and other antihyperglycemic medications on overall survival in diabetic patients with pancreatic cancer.

## 2. Material and Methods

The study was conducted in accordance with the Declaration of Helsinki, and the protocol was approved by the Ethics Committee of Vilnius Regional Biomedical Research Ethics Committee (Nr. 158200-17-913-423).

As this study was based on routinely collected administrative data, participant consent was not required. 

### 2.1. Data Sources

A retrospective cohort study was conducted using two different data sets: population-based Lithuanian Cancer Registry data, and healthcare service data from the National Health Insurance Fund (NHIF). Linkage was based on unique personal identification numbers, which are used throughout all information systems in Lithuania.

The Lithuanian Cancer Registry is a population-based database and covers all residents in Lithuania. The Cancer Registry database contains personal and demographic information (place of residence, sex, date of birth, vital status), as well as information on diagnosis (cancer site, date of diagnosis, method of cancer verification) and death (date of death, cause of death) of all cancer patients in Lithuania. The data quality, with regard to completeness and validity of case ascertainment, complies with international standards of cancer surveillance [32]. From this database, we obtained information regarding age at diagnosis, date of diagnosis, tumor classification (TNM), and cause and date of death for patients with pancreatic cancer.

Information regarding the diagnosis of T2DM and antihyperglycemic medications were obtained from the National Health Insurance Fund (NHIF) database. The National Health Insurance Fund (NHIF) database was created in 1999, seeking to reimburse healthcare institutions for the healthcare services provided from the NHIF. The system is used for the management, storage, exchange, analysis, and reporting of all the services provided by healthcare institutions. The national database contains demographic data and entries on the primary and secondary healthcare services provided, emergency and hospital admissions, and prescriptions of reimbursed medications. Data from the Lithuanian NHIF database encompass about 98% of inpatient cases and 90% of outpatient visits (up to 100% of primary healthcare visits) in Lithuania, covering the entire territory of the country [33]. The NHIF database contains reimbursed medications records. All dispensed medications are coded according to the Anatomical Therapeutic Chemical (ATC) medication classification, and the records include information on type of product, date, and quantity.

### 2.2. Study Population

The study population included diabetic patients with pancreatic cancer (ICD-10 C25) diagnosed between 1 January 2000 and 31 December 2015 (7653 cases). Cancer registry data were linked to the NHIF records on antihyperglycemic medication prescriptions (1009 cases). Patients with zero follow-up (156 cases), patients with less than 7 prescriptions of antihyperglycemic medications (310 cases), and those with a diagnosis of diabetes after cancer diagnosis (90 cases) were excluded from the analysis.

### 2.3. Exposure

In order to examine the impact of selected glucose-lowering therapies, subjects with T2DM were divided by selected regimens into six groups: sulfonylurea monotherapy; metformin monotherapy; insulin monotherapy; metformin and sulfonylurea combination; metformin and other antihyperglycemic medication combinations; all other combinations of oral antihyperglycemic medications. 

For further analysis of the association between metformin pancreatic cancer survival, patients were categorized according to metformin use (yes or no). All patients treated with metformin were categorized as metformin users, whereas use of medications other than metformin, including sulfonylurea, thiazolidinedione, insulin, or other oral antihyperglycemic medications, were categorized as non-metformin users. Cumulative doses of metformin were calculated from prescription data.

### 2.4. Diabetes Treatment Guidelines

According to Lithuanian diabetes management guidelines, metformin is the frontline preferred initial oral antihyperglycemic agent after diagnosis of T2DM. If metformin is contraindicated or intolerable as the initial treatment, then another class of antihyperglycemic agent can be used, depending on the clinical situation. If metformin monotherapy fails to achieve the glycemic goal (HbA1c less than 7%), the treatment is intensified adding a second-line medication, which is generally a sulfonylurea. As a third step, combination therapy with more than two classes of antihyperglycemic agents is used. Insulin treatment can be initiated after failed combination therapy or at any time, depending on the clinical situation. Metformin treatment is usually continued together with insulin [34].

### 2.5. Statistical Analysis

The primary outcome measure was all-cause mortality. Survival was calculated from the date of cancer diagnosis to the date of death or the end of follow-up (31 December 2018).

The chi-square test of independence was used to analyze the differences between groups. Univariate analyses using the Kaplan–Meier method examined the association of overall survival with age, gender, stage of disease, and glucose-lowering therapies. Survival curves were compared using the log-rank test. Multivariate Cox proportional hazards models were used to account for differences in cohort characteristics. The reference group was the sulfonylurea users group for all analyses. The threshold for statistical significance was set at the conventional level of α = 0.05, and 95% CIs for hazard ratios (HRs) were calculated.

All statistical analyses were carried out using STATA 15 statistical software (StataCorp. 2009. Stata Statistical Software: Release 15.0. College Station, TX, USA).

## 3. Results

Final study groups included 454 pancreatic cancer patients. During follow-up, there were 441 deaths including 406 from pancreatic cancer. Demographic and clinical characteristics of pancreatic cancer patients are presented in Table 1. There were no significant differences between the groups by gender, stage of the disease, and histology.

Kaplan–Meier survival analysis by age, gender, stage of the disease, and glucose-lowering therapies showed strong evidence of survival difference between the groups. In the multivariate analysis, risk differences in overall survival between patients by glucose-lowering therapy groups were not statistically significant (Table 2). However, the highest mortality risk was observed in patients on insulin monotherapy, and better survival was observed in the groups of patients using metformin, other antihyperglycemic medication combinations, and metformin and sulfonylurea combination therapy.

Analysis of survival between metformin users and non-users showed better survival in the metformin users group (*p* = 0.003) (Figure 1). As the group of never metformin users included pancreatic cancer patients on insulin monotherapy, survival between the sulfonylurea only and metformin only users was compared (Figure 2). Better survival in the metformin users group was observed (*p =* 0.014). In the multivariate analysis, after adjustment for sex, age, stage at diagnosis, and histology, metformin users’ risk of death remained lower but became insignificant (Table 3). Analysis by cumulative dose of metformin showed significantly lowest risk of death in the highest cumulative dose category (HR 0.76, 95% CI 0.58–0.99). Insignificantly lower risk of death was found in metformin users compared to sulfonylurea users (HR 0.74, 95% CI 0.54–1.02).

## 4. Discussion

Our study showed that better survival in diabetic pancreatic cancer was observed in the groups of patients using metformin, dual therapy of metformin and antihyperglycemic medication combinations, or metformin and sulfonylurea combinations.

The relationship between pancreatic cancer and diabetes mellitus is well known. T2DM is a known risk factor for pancreatic cancer [2]. The pathophysiologic derangements that are responsible for the development of diabetes mellitus have also been associated with an increased risk for cancer development [23].

There are a few studies showing that metformin use for T2DM treatment did not improve clinical outcomes in pancreatic cancer patients [29,30,31,35,36,37]. On the other hand, there are some large population studies implying that metformin can improve survival outcomes in pancreatic cancer [27,38,39,40]. This difference in the studies’ results may be explained by differences in patients’ demographic data, as the sample populations in those studies of better outcomes tend to be younger compared with studies showing no effect of metformin on pancreatic cancer.

Toriola et al. assessed a large population cohort of 3811 patients with pancreatic cancer (the vast majority being males) [35] and found no benefit of metformin to overall survival (OS) (HR = 1.05; 95% CI, 0.92–1.14; *p* = 0.28). However, a study showed that metformin use improved overall survival in non-Hispanic white patients (HR = 0.78; 95% CI, 0.61–0.99; *p* = 0.04) naive to metformin at initial pancreatic cancer diagnosis. Chaiteerakji and colleagues, using the time-varying Cox model, analyzed data of 980 pancreatic cancer patients with diabetes mellitus and found that metformin use was not associated with any survival benefit [29]. Additionally, two randomized trials failed to show any benefit of metformin in pancreatic cancer patients [30,31]. Reni and colleagues performed a phase II clinical trial including 60 randomly assigned patients and showed no benefit of metformin addition to a standard systemic therapy with cisplatin, epirubicin, capecitabine, and gemcitabine in patients with metastatic pancreatic cancer [30]. Another placebo-controlled trial by Kordes et al. also suggested that metformin use did not improve overall outcomes in pancreatic cancer. However, the sample size was also very small (N = 121), and patients tended to have an advanced stage of pancreatic cancer [31]. 

To the contrary, Amin et al., in a propensity score analysis [28] of a large cohort from the USA, found that metformin had a positive effect on survival compared to other antihyperglycemic agents’ use in pancreatic cancer patients. We also assessed various antihyperglycemic agents and their effect on survival and found that patients using insulin had the highest risk of death. This might be supported by in vitro shown mechanisms of metformin action [5,6,7,8,9,10,11,12,13,14,15,16,17]. Jang et al. analyzed a rather homogenous cohort of 764 pancreatic cancer patients with T2DM who underwent curative pancreas resection. Study results showed that patients with resectable pancreatic cancer and diabetes receiving metformin had lower cancer-specific mortality [38]. Sadeghi et al. analyzed 302 patients with diabetes and pancreatic cancer treated at The University of Texas MD Anderson Cancer Center, and the two-year survival rate was 30.1% for the metformin group and 15.4% for the non-metformin group (*p =* 0.004; χ(2) test) [40]. The median overall survival time was 15.2 months for the metformin group, and 11.1 months for the non-metformin group (*p =* 0.004, log-rank test). Metformin users had a 32% lower risk of death; the HR (95% confidence interval) was 0.68 (0.52–0.89) in a univariate model (*p =* 0.004), 0.64 (0.48–0.86) after adjusting for other clinical predictors (*p =* 0.003), and 0.62 (0.44–0.87) after excluding insulin users (*p =* 0.006). Metformin use was significantly associated with longer survival in patients with non-metastatic disease only [40]. In another small, retrospective study by Choi et al. investigators analyzed pancreatic cancer patients with T2DM receiving palliative chemotherapy and various antihyperglycemic agents [41]. Study showed better OS in diabetic patients (HR, 0.788; *p =* 0.059) and significant improvement of OS in the metformin treatment group in comparison within diabetes mellitus patients (HR 0.693; *p =* 0.036). 

Recently, at least five meta-analyses assessing metformin effects on survival in patients with pancreatic cancer and T2DM have been published [42,43,44,45,46]. These meta-analyses, including from 8 to 17 studies, concluded that metformin had a positive effect on pancreatic cancer patient survival. Li et al., in their meta-analysis, included nine retrospective studies and two randomized controlled trials [46]. In the subgroup analysis, authors found that that metformin improved survival in patients with resectable and locally advanced tumors, but not in patients with metastatic tumors. All these studies had common limitations: inclusion of retrospective cohorts, small sample sizes, various durations and exposure to metformin between the studies, and different characteristics of pancreatic cancer and treatments. Additionally, one meta-analysis assessed effects of a few antihyperglycemic agents (including metformin, sulfonylureas, thiazolidinediones, and insulin) to survival [47]. The meta-analysis included 14 studies, and only three of them additionally assessed effects of other medication than metformin [41,48,49]. They did not find any possible effect on survival in patients treated with medications other than metformin. 

Our study had several strengths. First, this was a large population-based study assessing possible effect of all of the most commonly used antihyperglycemic agents on survival in patients with pancreatic cancer and T2DM. Only three previous similar studies assessed all possible antihyperglycemic agents [41,48,49].

Our study had some limitations. First of all, this was a retrospective study with no evaluation of confounding factors, including body mass index, obesity, smoking history, lifestyle, dietary habits, or treatment modalities of the pancreatic cancer as these data were unavailable. Secondly, the results might be affected by the time-varying nature of antihyperglycemic medication use over time—the immortal time bias. Then, the number of cases in some groups was small; therefore, results of our study should be interpreted with caution. Finally, patients with other antihyperglycemic medications and with several agents used are prone to have more advanced diabetic disease, with possibly more comorbidities and worse survival, especially in insulin users group.

## 5. Conclusions

In conclusion, our study showed that metformin might have a benefit for survival of patients with pancreatic cancer, suggesting a potentially available option for the treatment. Further prospective studies and randomized controlled trials are needed to confirm these findings with the consideration of some confounding variables.

## Figures and Tables

**Figure 1 ijerph-17-06016-f001:**
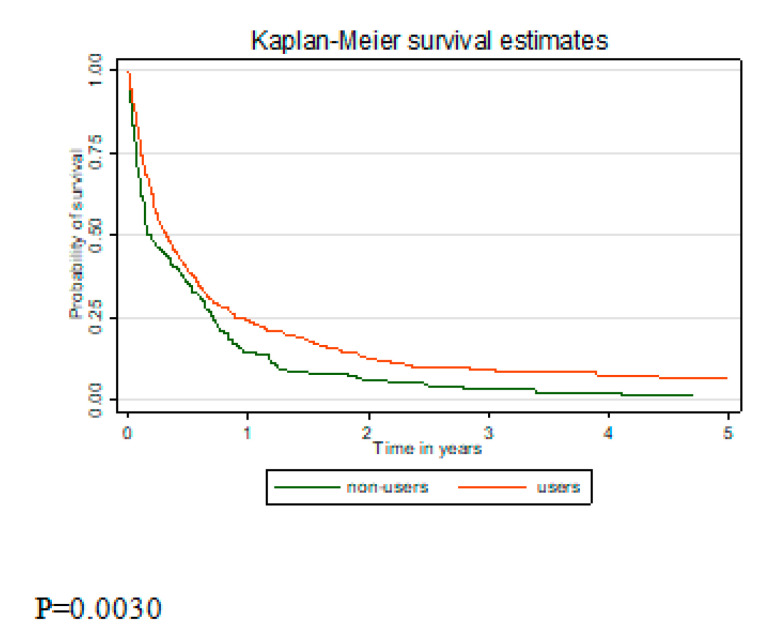
Kaplan–Meier survival curve comparing overall survival between diabetic metformin users and metformin non-users.

**Figure 2 ijerph-17-06016-f002:**
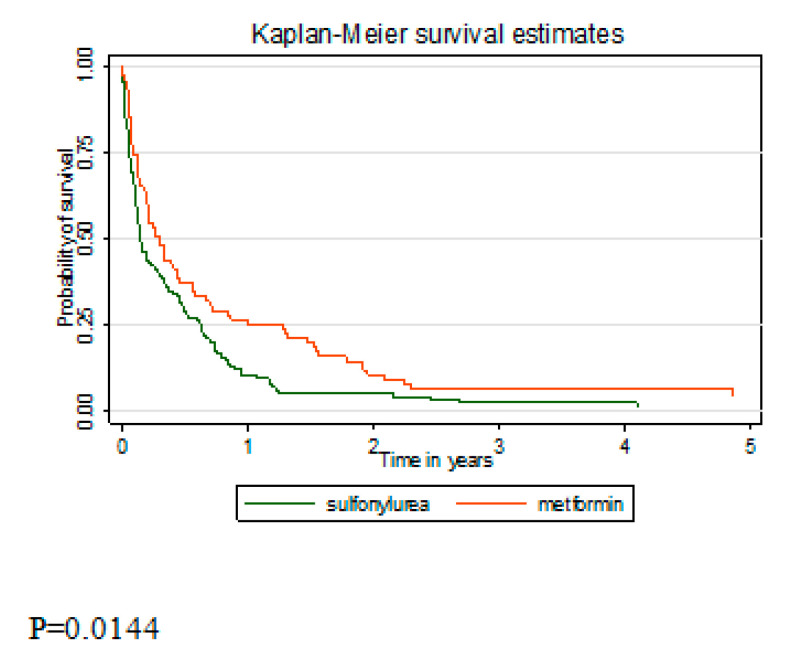
Kaplan–Meier survival curve comparing overall survival between diabetic sulfonylurea users and metformin users (monotherapy groups).

**Table 1 ijerph-17-06016-t001:** Demographic and clinical characteristics of pancreatic cancer patients by glucose-lowering therapies.

	Antihyperglycemic Medications						
Variable	Sulfonylurea	Metformin	Sulfonylurea and Metformin	Metformin and Other Medications	Insulin	Other Medications	*p* Value
Total	114	75	155	67	26	17	
Gender							
Male	49 (43.0)	33 (44.0)	60 (38.7)	32 (47.8)	13 (50.0)	8 (47.1)	0.78
Female	65 (57.0)	42 (56.0)	95 (61.3)	35 (52.2)	13 (50.0)	9 (52.9)	
Age at diagnosis							
<60	6 (5.3)	8 (10.7)	21 (13.5)	7 (10.4)	6 (23.0)	3 (17.7)	0.009
60–69	26 (22.8)	16 (21.3)	43 (27.7)	28 (41.8)	8 (30.8)	9 (52.9)	
70–79	48 (42.1)	36 (48.0)	62 (40.0)	22 (32.8)	8 (30.8)	4 (23.5)	
80+	34 (29.8)	15 (20.0)	29 (18.7)	10 (14.9)	4 (15.4)	1 (5.9)	
TNM stage							
I	4 (3.5)	5 (6.7)	11 (7.1)	3 (4.5)	3 (11.5)	1 (5.9)	0.94
II	16 (14.0)	12 (16.0)	26 (16.8)	15 (22.4)	4 (15.4)	5 (29.4)	
III	17 (14.9)	13 (17.3)	21 (13.5)	10 (14.9)	2 (7.7)	3 17.6)	
IV	61 (53.6)	33 (44.0)	72 (46.5)	28 (41.8)	14 (53.9)	7 (41.2)	
Missing	16 (14.0)	12 (160)	25 (16.1)	11 (16.4)	3 (11.5)	1 (5.9)	
Histology							
Adenocarcinoma	49 (43.0)	43 (57.3)	86 (55.5)	42 (62.7)	13 (50.0)	12 (70.6)	0.17
Other	12 (10.5)	3 (4.0)	6 (3.9)	5 (7.5)	2 (7.7)	1 (5.9)	
Neuroendocrine	4 (3.5)	1 (1.3)	10 (6.4)	1 (1.5)	2 (7.7)	1 (5.9)	
Not specified	49 (43.)	28 (37.4)	53 (34.2)	19 (28.3)	9 (34.6)	3 (17.6)	

**Table 2 ijerph-17-06016-t002:** Hazard ratio (HR) and 95% Confidence Interval (CI) of the association of glucose-lowering therapies, and overall mortality in diabetic pancreatic cancer patients.

Variable	Multivariate-Adjusted HR * (95% CI)	*p*-Value
Gender		
Male	1.00	ref.
Female	0.86 (0.70–1.05)	0.13
Age at diagnosis		
<60	1.00	ref.
60–69	1.02 (0.72–1.43)	0.91
70–79	1.44 (1.02–2.04)	0.04
80+	2.07 (1.40–3.06)	<0.001
TNM stage		
I	1.00	ref.
II	0.99 (0.62–1.59)	0.97
III	1.43 (0.88–2.31)	0.15
IV	2.80 (1.81–4.35)	<0.001
Missing	1.30 (0.80–2.10)	0.29
Histology		
Adenocarcinoma	1.00	ref.
Other	0.91 (0.60–1.37)	0.66
Neuroendocrine	0.27 (0.15–0.48)	<0.001
Not specified	1.26 (1.01–1.58)	0.04
Antihyperglycemic medications		
Sulfonylurea	1.00	ref.
Metformin	0.80 (0.59–1.09)	0.15
Sulfonylurea and metformin	0.79 (0.61–1.02)	0.08
Metformin and other medications	0.94(0.68–2.29)	0.69
Insulin	1.31 (0.83–2.04)	0.24
Other medications	0.61 (0.36–1.03)	0.07

* Adjusted for all variables shown in table.

**Table 3 ijerph-17-06016-t003:** HR and 95% CI of the association of metformin use, and overall mortality in diabetic pancreatic cancer patients.

Variable	Multivariate-Adjusted HR * (95% CI)	*p*-Value
By metformin use		
Non-users	1.00	ref.
Users	0.85 (0.70–1.04)	0.12
By metformin cumulative dose		
Non-users	1.00	ref.
<920.000	0.94 (0.72–1.21)	0.62
920.000–1.840.000	0.87 (0.67–1.12)	0.28
>1.840.000	0.76 (0.58–0.99)	0.04
By antihyperglycemic medication **		
Sulfonylurea	1.00	ref.
Metformin	0.74 (0.54–1.02)	0.07

* adjusted for gender, age group, stage of pancreatic cancer, and histology; ** monotherapy groups only.

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
