# Peer review of "Cohort Study of Antihyperglycemic Medication and Pancreatic Cancer Patients Survival"

_ijerph, 2020, doi:10.3390/ijerph17176016_

Round 1
Reviewer 1 Report
This is a well written manuscript aimed at investigating the association of the use of metformin and other antihyperglycemic medications on overall survival in diabetic patients with pancreatic cancer using data from a population-based cohort of cases registered between 2000 and 2015. This question is of interest and still discussed.
Since the access to administrative data sources (such as National Health insurance) and thus to the linkage with population-based cohorts, many studies have been set up to furnish complementary information regarding the question of treatment effectiveness or of treatment comparisons. In this context, researches in statistical methodology are currently discussed, developed and improved In order to supply (more or less) the absence of randomization and to control (more or less) for indication bias.
In the context of diabetic patients and metformin, one need to keep in mind that two recent randomized controlled trials failed to exhibit any influence of metformin on survival in diabetic patients with pancreatic cancer.
A Dulskas et al, conducted a well-designed, serious and clearly exposed study to investigate this question using population-based Lithuanian Cancer Registry data and health-care service data.
Yet, they honestly discussed the major limitations of their findings: no cofounding factors, immortal time bias, uncontrolled indications bias.
I would add that the number of cases in some groups is really too small to make sensed comparisons (for example, 26 patients with Insulin, or adjusting for histology in patients with metformin).
Minor remark
I questioned the availability and the repartition of TNM stage in the database concerning a cancer with around 1 patient on 2 not operated on. Are the "unclassified" (cases not resected and presenting without distant mestastasis)
The mention to cancer registry and national health care data should be added in the abstract
In conclusion, I find this study clear and honest, well presented and discussed. In the current context of ‘bigdata’, it seems to me that it must be highlighted. The results must be taken with caution due to the real limits of the study. That is the reason why I would recommend to publish it, but without “high priority”.
Author Response
Dear Reviewer,
Thank you for your letter and constructive comments concerning our manuscript entitled “Cohort study of antihyperglycemic medication and pancreatic cancer survival”. The paper was revised substantially. Following changes have been made. They are as follows:
Revised paragraphs, sentences, words are below:
This is a well written manuscript aimed at investigating the association of the use of metformin and other antihyperglycemic medications on overall survival in diabetic patients with pancreatic cancer using data from a population-based cohort of cases registered between 2000 and 2015. This question is of interest and still discussed.
Since the access to administrative data sources (such as National Health insurance) and thus to the linkage with population-based cohorts, many studies have been set up to furnish complementary information regarding the question of treatment effectiveness or of treatment comparisons. In this context, researches in statistical methodology are currently discussed, developed and improved In order to supply (more or less) the absence of randomization and to control (more or less) for indication bias.
In the context of diabetic patients and metformin, one need to keep in mind that two recent randomized controlled trials failed to exhibit any influence of metformin on survival in diabetic patients with pancreatic cancer.
A Dulskas et al, conducted a well-designed, serious and clearly exposed study to investigate this question using population-based Lithuanian Cancer Registry data and health-care service data.
Yet, they honestly discussed the major limitations of their findings: no cofounding factors, immortal time bias, uncontrolled indications bias.
I would add that the number of cases in some groups is really too small to make sensed comparisons (for example, 26 patients with Insulin, or adjusting for histology in patients with metformin).
Thank you for the comments. The suggested limitation added to “Limitations” section.
Minor remark
I questioned the availability and the repartition of TNM stage in the database concerning a cancer with around 1 patient on 2 not operated on. Are the "unclassified" (cases not resected and presenting without distant mestastasis)
In this study we used TNM stage as reported to the Cancer Registry. Unfortunately, no information on treatment methods is available in the Cancer Registry database. Proportion of unclassified tumours should be interpreted as indicator of the quality of Cancer Registry data, not as proportion of not operated cases.
The mention to cancer registry and national health care data should be added in the abstract
Cancer registry and national health care data added in the abstract.
In conclusion, I find this study clear and honest, well presented and discussed. In the current context of ‘bigdata’, it seems to me that it must be highlighted. The results must be taken with caution due to the real limits of the study. That is the reason why I would recommend to publish it, but without “high priority”.
Thank you for your comment.
Thank you very much indeed.
Sincerely Audrius Dulskas
Reviewer 2 Report
In this study, the authors investigated the possibility of improving the survival rate with antihyperglycemic medication, such as metformin, in diabetes patients with pancreatic cancer. They argued that metformin might lower the risk of death in patients with pancreatic cancer, but there are some issues to address. 1. Pancreatic cancer is very closely related to the incidence of type 2 diabetes, and 80% of pancreatic cancer patients develop type 2 diabetes. Therefore, the argued that patients receiving Insulin are at higher risk of death is a very hasty conclusion. This is because pancreatic cancer itself has a high mortality rate. The authors are encouraged to revise the theoretical background and results. 2. In 2017, meta-analysis results for clinical use of metformin based on pancreatic cancer patients were published (Scientific Reports volume 7, Article number: 5825). In this study, the author concludes: "The later the tumor stage, the more obscure the effect of metformin with respect to its efficacy." In this study, the authors performed an analysis on patients with relatively advanced stages. In Figures 2 and 3, if the values for median survival are presented, it anticipates that there is no significant change between the cohorts. 3. There are many typos and grammatical errors.
Author Response
Dear Reviewer,
Thank you for your letter and constructive comments concerning our manuscript entitled “Cohort study of antihyperglycemic medication and pancreatic cancer survival”. The paper was revised substantially. Following changes have been made. They are as follows:
Revised paragraphs, sentences, words are below:
In this study, the authors investigated the possibility of improving the survival rate with antihyperglycemic medication, such as metformin, in diabetes patients with pancreatic cancer. They argued that metformin might lower the risk of death in patients with pancreatic cancer, but there are some issues to address.
- Pancreatic cancer is very closely related to the incidence of type 2 diabetes, and 80% of pancreatic cancer patients develop type 2 diabetes. Therefore, the argued that patients receiving Insulin are at higher risk of death is a very hasty conclusion. This is because pancreatic cancer itself has a high mortality rate. The authors are encouraged to revise the theoretical background and results.
We completely agree with reviewer comments, that most of the patients with pancreatic cancer eventually will develop diabetes and that pancreatic cancer has a high mortality rate. However, aim of this study was to assess the association between the use of on overall survival in diabetic patients with pancreatic cancer.
Higher mortality risk in insulin users probably related to severity of diabetes and the high mortality rates not surprisingly are higher in this group of patients. As our primary aim was to analyse influence of metformin on pancreatic patient survival, we did not discuss separately insulin.
We slightly modified results section in order to meet reviewer comments
2. In 2017, meta-analysis results for clinical use of metformin based on pancreatic cancer patients were published (Scientific Reports volume 7, Article number: 5825). In this study, the author concludes: "The later the tumor stage, the more obscure the effect of metformin with respect to its efficacy." In this study, the authors performed an analysis on patients with relatively advanced stages. In Figures 2 and 3, if the values for median survival are presented, it anticipates that there is no significant change between the cohorts.
Suggested meta-analysis included in the discussion.
- There are many typos and grammatical errors.
Typos and grammatical errors corrected.
Thank you very much indeed.
Sincerely
Audrius Dulskas, MD, PhD
Round 2
Reviewer 2 Report
All concerns have been well addressed. There is no additional comment to raise.